# Fibrin-Rhamnogalacturonan I Composite Gel for Therapeutic Enzyme Delivery to Intestinal Tumors

**DOI:** 10.3390/ijms24020926

**Published:** 2023-01-04

**Authors:** Dzhigangir Faizullin, Yuliya Valiullina, Vadim Salnikov, Pavel Zelenikhin, Yuriy Zuev, Olga Ilinskaya

**Affiliations:** 1Kazan Institute of Biochemistry and Biophysics, FRC Kazan Scientific Center of RAS, Kazan 420111, Russia; 2Institute of Fundamental Medicine and Biology, Kazan Federal University, Kazan 420008, Russia

**Keywords:** drug delivery, colorectal cancer, binase, fibrin, rhamnogalacturonan I

## Abstract

Therapy of colorectal cancer with protein drugs, including targeted therapy using monoclonal antibodies, requires the preservation of the drug’s structure and activity in the gastrointestinal tract or bloodstream. Here, we confirmed experimentally the fundamental possibility of creating composite protein–polysaccharide hydrogels based on non-degrading rhamnogalacturonan I (RG) and fibrin as a delivery vehicle for antitumor RNase binase. The method is based on enzymatic polymerization of fibrin in the presence of RG with the inclusion of liposomes, containing an encapsulated enzyme drug, into the gel network. The proposed method for fabricating a gel matrix does not require the use of cytotoxic chemical cross-linking agents and divalent cations, and contains completely biocompatible and biodegradable components. The process proceeds under physiological conditions, excluding the effect of high temperatures, organic solvents and ultrasound on protein components. Immobilization of therapeutic enzyme binase in the carrier matrix by encapsulating it in liposomes made from uncharged lipid made it possible to achieve its prolonged release with preservation of activity for a long time. The release time of binase from the composite carrier can be regulated by variation of the fibrin and RG concentration.

## 1. Introduction

Today, colorectal cancer is the third most common cancer worldwide. Despite advances in immune and targeted therapies, according to the National Cancer Institute, the five-year survival rate for patients with advanced colorectal cancer is only 11% [1]. The attention of researchers is focused on the search for new alternative approaches to therapy, and, in particular, the use of promising antitumor enzymes of the ribonuclease class, which demonstrate biological activities such as control of blood vessel growth, antiviral activity [2,3,4] and toxicity against tumor cells [5]. Binase, from *Bacillus pumilus*, is a representative of RNases that are phylogenetically distant from their counterparts in mammals and, therefore, are not affected by a tissue inhibitor of RNases in the human body. Binase has proven to be a promising agent that induces apoptosis in a number of tumor cells expressing *RAS*, *KIT*, and *AML/ETO* oncogenes [6,7] and also suppresses tumor metastasis in animals [8].

It is known that the transit time of substances from the oral cavity to the stomach is 0.5–2 h, the substances are in the small intestine for 1–4 h, and can remain in the large intestine for more than a day. This means that some carriers with immobilized binase, after a supposed single oral dose, have to provide the prolonged release of binase throughout the entire gastrointestinal tract, where, in the presence of tumors, the enzyme can exert its therapeutic effect. Recently, we found that binase loaded into hydrogel microspheres based on alginate and gelatin, fortified with calcium ions, gradually leaves them, causing apoptosis of human duodenal adenocarcinoma cells [9]. However, strengthening carriers with ions introduces an undesirable non-selective toxicity towards the glandular epithelium of the stomach and enterocytes of the intestine [10]. Therefore, the problem of creating biocompatible and biodegradable carriers that do not need stabilization with divalent ions is an urgent one.

Various constructs of drug carriers based on pectin polysaccharides are of interest, since pectin is resistant to the action of digestive enzymes of the upper gastrointestinal tract and can be degraded only by the action of microbial enzymes in the colon [11]. The experience of using anionic pectin polysaccharides for drug encapsulation has exhibited a number of advantages: ease of gel formation, biodegradability, and low cost [12]. However, charged polysaccharides are unstable in body fluids [13,14], and divalent cations usually used to stabilize pectin gels exhibit cytotoxicity.

Here, we set a goal to create a complex biocompatible and biodegradable carrier for an antitumor protein drug included in liposomes without the use of divalent ions. In this regard, we turned to such polymers as RG and fibrin. Hydrogels, based on polysaccharides, proteins or polysaccharide–protein combinations, display original structural, physicochemical [15,16,17,18] and medico-biological properties in drug delivery [19,20], tissue engineering and regenerative medicine [21], and biosensing and molecular diagnostics [22], therapy [23].

Rhamnogalacturonans, which are weakly charged components of pectin, had attracted little attention until their specific bioactivity was discovered. It has been shown, in particular, that potato RG inhibits the proliferation of certain types of cancer cells [24], stimulates the immune system, suppresses metastases, and exhibits prebiotic properties [25,26,27]. Potato RG differs from RG from other sources in its increased content of galactan side chains [28]. According to available data, it is the neutral galactan side chains of RG that determine its anticancer activity [29]. Of no small importance is the availability and low cost of the product [30], which makes it a promising component of artificial drug carriers for intestinal anticancer therapy.

The possibility of using fibrin clots as a drug carrier has been studied in a number of works [31,32,33]. It was shown that the drug retention time depends on the pore size, the resistance of the fibrin matrix to the external environment, and the interaction of the drug with fibrin. By varying the concentration of fibrinogen (Fb) and thrombin, one can influence the pore size and fiber thickness, and the rate of fibrin proteolysis. Resistance to external factors is achieved by creating protective shells or conjugates of fibrin with protective polymers [34]. Fibrin shows a pronounced affinity for hydrophobic surfaces; therefore, modification of drugs by introducing hydrophobic groups increases their retention time in the fibrin matrix [34]. This property of fibrin looks especially attractive in light of the creation of conjugates of fibrin and potato RG, due to the relatively lower hydrophilicity of galactan chains and their tendency to self-associate in an aqueous solution [35,36,37].

The only example of use of potato RG for drug encapsulation again required special gel strengthening methods [38], since the development of RG-based carriers is complicated by their weak gelling ability. Our proposed method for constructing a carrier matrix does not require the use of chemical cross-linking agents and contains completely biocompatible and biodegradable components. The method is based on enzymatic polymerization of fibrin in the presence of potato RG and liposomes with an encapsulated protein drug. Such a complex design allows for the stability and prolonged release of antitumor RNase.

## 2. Results

### 2.1. Morphology of Composite Gels

The coagulation ability of Fb strongly depends on external conditions, such as salinity, pH, temperature etc., and on the state of the Fb molecule itself. Even minor chemical modifications (for example, decialization and point mutations) or conformational alterations upon adsorption on surfaces or crowding would significantly affect clotting [17,39,40,41]. It can be expected that the presence of RG and liposomes in the reaction mixture would affect the structure of composite gels. According to the SEM images, the unmodified fibrin gel had a porous structure and consisted of crimped smooth weakly branching fibers, which were 104 ± 30 nm thick (Figure 1a). The structure of the Fb/RG composite gel (Figure 1b) looked denser compared to the control, formed by more straight and thinner fibers (83 ± 12 nm) with a large number of either spherical particles associated with fibrin fibers or branched start formations. The size of these formations was 10–15 nm, which was comparable to the hydrodynamic diameter of RG molecules in solution [37]. Pure RG looks like a scattering of individual particles or their agglomerates, without the formation of something like fibers (Figure 1d), which is consistent with the concept of globular conformation of RG in solution [36,37]. It can be assumed that in the composite gel, RG also bound to fibrin fibers in a globular conformation. In the Fb/RG/DPPC ternary system (Figure 1c), a dense gel was also formed with fibers thinner than in pure fibrin and showing a tendency to stick.

The fibers looked rougher than in Figure 1b, due to the greater number of particles bound to the surface or branch start formations. Thus, Fb retained the ability to gel in the presence of RG and liposomes from DPPC; however, the morphology of the composite gel and the structure of its fibers differed significantly from those of pure fibrin.

### 2.2. Fibrin Formation and Fibrinolysis in the Presence of Liposomes and RG

The influence of composite gel components on the process of fibrin formation and the resistance of the gel to hydrolysis was analyzed according to the dynamics of turbidity changes. In a fibrinogen solution, the addition of thrombin initiated the process of fibrin formation, which was accompanied by an increase in light scattering (Figure 2A). The growth of scattering began with a slight delay from the moment of thrombin introduction. During this time (lag period), the process of hydrolytic activation of fibrinogen and formation of short protofibrils occurred, which then combined into fibrils growing in the longitudinal and lateral directions, which was manifested by an increase in turbidity [39]. The plateauing of the turbidity curve indicated the completion of gel formation. The level of scattering achieved depended on the thickness of fibers and the density of the three-dimensional network of gel. In comparison with pure Fb, in the Fb/RG mixture, the kinetics of fibrin formation was characterized by an increased lag time and scattering on the plateau. Various factors, influencing diffusion, as well as fibrin monomer interactions, could be the reasons behind the observed nonlinear kinetic alterations (Figure 2A,B).

At the same time, the scattering on the plateau increased with RG concentration, which might indicate the formation of a denser gel in accordance with morphological data, and might also reflect changes in the internal structure of fibers [39,42]. The introduction of the third component, DPPC, into the system led to a reduction in the lag period to almost zero and increased the rate of fibrin formation (Figure 2A,B). These features of fibrin formation in the presence of DPPC have already been discussed previously [17]. They are associated with the local concentration of fibrinogen molecules at the surface of DPPC liposomes and their faster polymerization compared to molecules in the solution bulk. Fibrin associated with liposomes is included in the general spatial network of gel, which contributes, as shown below, to a better retention of liposomes in fibrin gel. Diffusion of protease trypsin into the Fb/RG composite gel was slower, which manifested itself in the delay of gel disintegration after the moment when trypsin was added. The delay depended on the RG concentration and increased rapidly in more concentrated RG solutions. However, the rate of clot lysis itself increased in this case (Figure 2C), which could be the result of modification of clot architecture (branching, size of fibers and extent of crosslinking), as well as of the structure of the internal fibers.

### 2.3. Interaction of Gel Components

The increased affinity of Fb for hydrophobic surfaces was clearly manifested in its interaction with the lipid bilayer. Figure 3A shows the sorption kinetics of Fb on a bilayers of DPPC and PC. Both lipids were electrically neutral and had identical head group structures. However, the presence of unsaturated fatty acid tails of various length in the PC resulted in a higher mobility of molecules in the bilayer and a higher hydration of their head groups [17,43], which significantly affected the interaction with Fb. It followed from the coefficients of two-state binding approximation of the presented data that the adsorption of Fb on the DPPC bilayer was a two-stage process, while on the PC one it was essentially a single-stage process. On the rigid and relatively weakly hydrated surface of the DPPC bilayer, most of the Fb was irreversibly bound, in contrast to the case of a friable and more hydrated PC surface, from which most adsorbed Fb molecules are washed off by buffer. The study of the liposomes’ spontaneous release from the fibrin gel matrix confirmed that DPPC-built liposomes were better retained than PC-built liposomes, due to stronger binding to Fb (Figure 3B).

Figure 4A shows the dependence of the zeta potential of the RG particles in solution on Fb concentration. RG particles themselves carry a negative potential of −13.7 ± 1.2 mV. When Fb is introduced, the zeta potential becomes more positive, and the dependence has the form of a curve with saturation, indicating the binding of RG and Fb molecules. The zeta potential already reached saturation at 0.125 mg/mL of Fb, increasing by 15%. Figure 4B shows the particle size distribution in the same solutions. A narrow symmetrical peak with a maximum at 8 nm was observed in the RG solution. According to the literature, RGs of similar structure from other sources have the conformation of spherical compact particles in solution. Their size varies from 4 to 80 nm depending on the RG structure, mainly on the composition and length of the arabino–galactan side chains, as well as on the polysaccharide concentration [37,44,45,46]. There was no reason to believe that the spatial structure of RG used would differ significantly from those described in literature. Let us assume, therefore, that RG in an aqueous solution exists in the form of spherical particles with a hydrodynamic diameter of 8 nm. With the addition of 0.062 mg/mL Fb, a tail appeared on the distribution from the side of large dimensions. Increasing the Fb concentration to 0.125 mg/mL caused a narrowing of distribution and a shift of the maximum to 11 nm. Subsequent portions of the protein again led to an increase in the distribution width and a gradual shift of the maximum to 18 nm. Taking into account that, according to our data, the hydrodynamic diameter of the Fb molecule was 20 nm, which matched the known data [47], it could be concluded that the formation of complexes was mainly completed at 0.125 mg/mL Fb, and the addition of Fb only led to an increase in concentration of free protein molecules. Figure 4C shows the proportions of particles with a size of 8 and 16 nm. The fitting of dependence for 16 nm size particles, according to the Hill equation, gave the value of the cooperativity parameter as 1.89 ± 0.59. This was a somewhat dubious value due to the unknown molecular weight of RG. However, this suggested a cooperative nature of binding.

Structural changes in the Fb molecule upon interaction with RG were studied by FTIR spectroscopy. Figure 5A shows the FTIR spectra during successive adsorption from solution, first with Fb and then with RG, washing with buffer after each stage. The spectra indicated the irreversible binding of RG to Fb. The difference spectrum (Figure 5A) illustrates the changes of spectral components as a result of binding. An increase in intensity in the 1715 cm^−1^ band indicated the protonation of a part of the carboxyl groups. In this case, there was a corresponding decrease of the ionized carboxyl absorption at 1566 and 1399 cm^−1^. These carboxyl groups obviously belonged to Fb, since at frequencies of 1590 and 1415 cm^−1^, coinciding with the absorption maxima of the ionized RG uronic acid residues, only an increase in intensity of the difference spectrum was observed. The intensive multicomponent band of C–O and C–C vibrations of polysaccharide groups in the region of 1000–1100 cm^–1^ did not undergo noticeable changes relative to the pure RG, which indicated that the conformation of these groups was retained during adsorption on Fb [35,48]. In the absorption region of Fb peptide groups, a narrowing of the amide1 band was observed with a slight intensity increase at 1652 cm^−1^ and a decrease at 1643 and 1670 cm^−1^, which was clearly seen in the second derivative plots (Figure 5B). These changes could be interpreted somewhat as a gain of helicity at the expense of unordered regions of the protein. The uronic acid residues of the RG main chain, apparently, did not participate in the interaction. The partial protonation of protein carboxyls might be the result of their dehydration upon binding with RG molecules. Thus, the IR spectra indicated that RG and Fb formed stable complexes without significant changes in their secondary structure.

Encapsulation of a protein in liposomes is a common and effective way to achieve its long-term stability in carriers [49,50]. Encapsulation can be accompanied by protein distribution, not only in the water core of the liposome, but also in the lipid bilayer, which can affect the release kinetics and catalytic properties of enzymes [51]. The interaction of binase with the DPPC bilayer was determined, measuring the half-transition temperature of turbidity in lipid dispersion (Table 1). Simple sigmoid curves were obtained in all cases.

The half-transition temperature determined in pure DPPC suspension was 40.3 °C, which was close to the phase transition temperature of the DPPC bilayer [52]. The transition temperature of DPPC in the presence of binase decreased by 0.84 ± 0.05 °C. This observed decrease in bilayer stability indicated the binding of binase.

### 2.4. Resistance of Composite Gel to pH

We studied the effect of gel composition on its stability in Tris–HCl solutions at pH 2 and 7.4, simulating the acidic and slightly alkaline environments of the stomach and the colon, respectively [53]. Figure 6A shows the values of Fb concentration in supernatant versus the initial Fb concentration in the gel after 24 h of incubation at respective pHs. Incubation of pure fibrin at pH 2 led to its fast and complete dissolution, due to decomposition into monomers. The rate of gel decomposition did not depend on Fb concentration and the process was basically completed after 3–4 h of incubation.

However, in a Fb/RG/DPPC composite gel, this process became more complex. Approximately 20–25% of fibrin dissolved in the same 3–4 h, but the remaining part was resistant to pH up to 24 h of observation. Visually, the gel in liquid remained intact and retained its original whitish-turbid color and shape. It could be assumed that the part of fibrin that remained stable in the acidic medium was formed by Fb molecules associated with RG. At pH 7.4, the destruction of pure fibrin sharply slowed down due to the restoration of knob-hole interactions between fibrin monomers, which are mainly electrostatic in nature [54]. At the same time, in the composite gel, under the same conditions, the stability increased so much that the concentration of fibrin dissolved in 24 h did not exceed the measurement error. Thus, due to the reinforcement of fibrin fibers with RG molecules, an increased resistance of composite gel to the action of pH in both slightly alkaline and acidic regions was achieved.

### 2.5. Release of Binase from Composite Gel

Figure 6B presents the kinetics of binase release from fibrin gels of various compositions at pH 7.4, measured spectrophotometrically. The release rate of binase immobilized in pure fibrin gel showed a strong dependence on fibrin concentration. The maximum release rate was observed in the gel with Fb concentration of 1 mg/mL. During the first 24 h, 80% of binase was released, and in 48 h, almost all the binase had gone into the solution. With an increase in the concentration of fibrin, the release rate rapidly decreased, and no more than 20% of enzyme was released from gel with 5 mg/mL of fibrin into the surrounding solution in 24 h. Even more so, the release of binase slowed down by its inclusion into the lipid vesicles immobilized in the fibrin gel. In a gel of 5 mg/mL Fb with DPPC, only about 10% of encapsulated enzyme was released in 24 h. A qualitatively similar picture was observed when measuring the activity of binase released from pure fibrin and from liposomes embedded in fibrin gel (Figure 6C). However, the reinforcing of fibrin gel with RG had a truly cardinal effect. In the Fb/RG/DPPC composite gel, binase was retained so efficiently that it was not possible to reliably measure the amount of released enzyme in 24 h.

### 2.6. Cytotoxic Effect of Binase towards Cancer Cells

It was found that composite gel matrices that do not contain binase were not toxic towards *HuTu80* cells, and the distribution of cell populations in the respective samples was identical to the negative control (Figure 7). The average survival of cells in the positive control (binase in solution) was 74.1%. In the case of matrices loaded with binase, the proportion of non-apoptotic cells was 86.6 ± 1.07% and 84 ± 0.8% for Fb and Fb/DPPC, respectively, while the ratio of late-apoptotic cells to early-apoptotic cells was 1:1.4 in in the first case and 1:1.5 in the second case. Immobilization of binase in Fb and Fb/DPPC changed the ratio of cells in early and late apoptosis, increasing the proportion of early apoptotic cells. The result obtained was a consequence of prolonged action of immobilized binase and gave evidence that the immobilized enzyme did not lose its pro-apoptotic effect compared to the pure enzyme.

## 3. Discussion

Fibrin in nanomedicine is considered a promising material for drug delivery and nanoprosthetics, due to its absolute biocompatibility and biodegradability, good tissue adhesion and adsorption capacity. At the same time, the use of fibrin requires special attempts to regulate its resistance to tissue fluids and the release of drugs, for which various composites are widely used. In this work, we chose a similar approach to create a carrier targeted to the large intestine. For this purpose, we used the complexing of fibrin with modified potato rhamnogalacturonan I. The latter is known to resist the fluids of the upper gastrointestinal tract and is only cleaved by the colon microbiota [38]. The resulting fibrin–polysaccharide composite gel is convenient to use, as it allows implantation in a liquid form, followed by in situ hardening, but can also be used orally in the form of preformatted capsules, since it showed high resistance to the acidic pH of the stomach. The absence of a toxic effect on normal cells of its ingredients, these being fibrin and rhamnogalacturonan I (potato galactan), as well as DPPC liposomes, was previously established [11,31,32].

As an antitumor agent, we used the bacterial RNase binase from *Bacillis pumilus*, which previously showed inhibition of the proliferation and migration of intestinal cancer cells [55]. We previously showed that binase did not act on normal fibroblasts but had a toxic effect on ras-transformed tumor cells [6]. The safety of RNases application was confirmed. For example, RNase is used as a commercial antiviral drug registered in the Vidal reference book (https://www.vidal.ru/drugs/ribonuclease__24326; accessed on 3 January 2022).

To protect binase from the action of the gastrointestinal tract environment and prolong its release from the carrier matrix, we used encapsulation of the enzyme in liposomes formed from a neutrally charged lipid, followed by immobilization of liposomes in a composite gel. The creation of such a complex construct required a comprehensive study of the interaction of its components with each other, for which various physicochemical methods were used.

With SEM, it was shown that the morphology of the composite gel retained the main features of pure fibrin, being characterized by a denser spatial network of fibers with submicron pores (Figure 1). The kinetics of fibrin polymerization in the presence of RG differed from the control by a high level of scattering on the plateau (Figure 2A), the value of which increased rapidly with increasing RG concentration, indicating the formation of a denser gel structure and/or thicker fibers [39]. Micrographs (Figure 1) showed that RG was adsorbed on the surface of fibrin fibers in the form of globular particles. At the same time, it followed from the FTIR spectroscopy data that excess RG was removed from the fibrin gel during washing, while the other part remained tightly bound to fibrin, without significantly affecting its secondary structure (Figure 5). The study of the Fb and RG binding in solution showed that the stoichiometric complex was formed at a weight ratio of Fb/RG equal to 0.125 (Figure 4).

The binding of RG to fibrin fibers significantly increased the stability of composite gel. A pure fibrin gel undergoes rapid disintegration at pH2, while only about 20–25% of the initial amount was released from the composite in 24 h.

Despite the significant pore size, the composite retained lipid particles well due to the high affinity of fibrin to the surface of DPPC vesicles (Figure 3). In previous studies [17], we showed that fibrin/fibrinogen binding on the surface of lipid bilayer depended on its hydrophobicity. Neutrally charged lipids with saturated fatty acid tails, including DPPC, formed a densely packed bilayer, the surface of which was relatively weakly hydrated and adsorbed hydrophobic molecules well.

Binase contains hydrophobic patches on its surface, especially the hydrophobic region of α-helix II on the N-terminus [56], that allows it to bind to hydrophobic surfaces. As evidenced by a decrease in the DPPC phase transition temperature (Table 1), during encapsulation of the enzyme in the aqueous pool of liposomes, a part of the enzyme binds to the lipid bilayer.

The encapsulated binase in the composite gel was characterized by a long retention time (Figure 6B), slowly releasing from the gel, and exhibited an apoptotic effect towards duodenum cancer cells (Figure 7). For binase and its homologue barnase, the enzymes were shown to be stable over a wide pH range from 3 to 10 [57], which allowed the bacterial enzyme to maintain catalytic activity in the intestine.

It is known that the *RAS* oncogene is among the most mutated in cancer. *RAS* mutations are identified in about half of patients diagnosed with colorectal cancer, conferring poor prognosis and lack of response to the anti-EGFR (epidermal growth factor receptor) therapy [58]. Direct inhibition of mutated *RAS* is one of the most promising strategies in colorectal cancer, as well as in other solid malignancies. Analyzing patients with colorectal cancer in the Republic of Tatarstan (Russian Federation), we found that the total mutation rate of *KRAS* was about 57% and the frequency of mutant *KRAS* occurrence in patients aged over 60 years decreased sharply, compared with younger ones, and did not exceed 10% [59]. This means that young people who do not benefit from anti-EGFR therapy especially need the use of agents that block mutant *RAS*, such as binase [7], the prolonged action of which we demonstrated by inclusion in the composite gel.

## 4. Materials and Methods

### 4.1. Chemicals

Fibrinogen from bovine plasma (Fb) (341573, Calbiochem, Darmstadt, Germany); bovine plasma thrombin (Tb) (T4648, Sigma-Aldrich, Taufkirchen, Germany); trypsin from porcine pancreas (T0303, Sigma-Aldrich, Taufkirchen, Germany); modified potato RG (Lot 120501c, Megazyme, Wicklow, Ireland) were used to create a composite carrier. The RG we used was a product of enzymatic hydrolysis of potato rhamnogalacturonan I and consisted of a backbone with alternating galacturonic acid and rhamnose residues and side chains, predominantly galactan, of various lengths. In comparison with raw rhamnogalacturonan I, this product did not contain extended sections of galacturonic acid, that significantly reduce the polymer overall charge. In addition, such a galactan-enriched product was easier to ferment in the colon [28]. The chemical composition of the modified potato RG, according to the manufacturer’s specification, included 87% galactose, 3% arabinose, 4% rhamnose, and 6% galacturonic acid and was close to that used in [38].

Liposomes prepared from phosphatidylcholine (PC) isolated from egg yolk lecithin (8640, Sigma-Aldrich, Taufkirchen, Germany) and 1,2-dipalmitoyl-sn-glycero-3-phosphocholine (DPPC) (850355, Avanti Polar Lipids, Alabaster, AL, USA) were used as the lipid component.

Stock solutions of Fb and RG were prepared either in high-salt buffer 1 (20 mM Tris-HCl with 150 mM NaCl, pH 7.4) or in low-salt buffer 2 (5 mM Tris-HCl with 30 mM NaCl, pH 7.4); thrombin and trypsin were dissolved in buffer 1.

### 4.2. Enzyme

The guanyl-specific RNase from *Bacillus pumilus* 7P, binase, (12.2 kDa, 109 amino acid residues, pI 9.5) was isolated from the culture fluid of *Escherichia coli* BL21 carrying the *pGEMGX1/ent/Bi* plasmid, according to Dudkina et al. [59]. The catalytic activity against synthetic substrates [60], and high-polymeric yeast RNA was already known [61,62]. A maximum activity, determined according to Kolpakov and Ilinskaya [63], was 14,000,000 U/mg at pH 8.5. An activity unit is the amount of enzyme capable of increasing the extinction at 260 nm of acid-soluble products of RNA hydrolysis to 1 unit per min at 37 °C and pH 8.5.

### 4.3. Preparation of Solutions

Commercial lyophilized Fb powder was dissolved in 0.9% aqueous NaCl at 37 °C and transferred to buffer 1 or buffer 2 using Zeba 7K MWCO desalting columns (Thermo Fisher Scientific, Waltham, MA, USA). The protein concentration was determined using extinction coefficients E^0.1%^_280nm_ for Fb: 1.51 [64], binase: 2.2 [9], trypsin: 1.6 [65]. Tb, at a concentration of 113 NIH units/mg, was dissolved in buffer 1. Stock solutions of Fb and Tb were aliquoted, frozen in liquid nitrogen and stored at −18 °C until use. Stock solutions of trypsin at a concentration of 1 mM, binase (14 mg/mL), and RG (10 mg/mL) were prepared immediately before measurements.

Fibrin coagulation was performed by the introduction of Tb, but without the addition of external calcium, in order to exclude the toxic effect of Ca^2+^ ions on the cells under study.

### 4.4. Preparation of Liposomes

Weighed portions of DPPC and PC were dissolved in chloroform. Chloroform was evaporated from lipid emulsion until a thin lipid film was formed. The film was diluted with buffer 1. The resulting coarse lipid dispersion, with a lipid concentration of 4 mg/mL, was subjected to 3 cycles of freezing (at liquid nitrogen temperature) and thawing (+55 °C). A stock solution of binase (experiment), or an equivalent amount of buffer (control), was added to the suspension of liposomes, after which it was repeatedly extruded at +55 °C through polycarbonate filters (Avanti Polar Lipids, Alabaster, AL, USA) with a pore size of 100 nm. The suspension with binase was washed away from the protein not bound to liposomes by centrifuging three times at 15,000× *g* for 30 min and replacing the supernatant with fresh buffer. Protein concentration in the suspension was determined spectrophotometrically at a wavelength of 280 nm with a control suspension of DPPC placed in a reference cuvette. Care was taken to ensure that the total absorbance at 280 nm, due to liposome scattering and protein absorption, did not exceed 0.5 in both the measurement and reference cuvettes [66]. Besides, the actual content of binase in the liposomes was verified by measuring the protein concentration in the washing supernatants. The results obtained by the two methods agreed with each other within 10%.

### 4.5. Microgravimetry

Measurements were carried out using a QCM-200 quartz microbalance (SRS, Sunnyvale, CA, USA). We used 5 MHz quartz resonators with gold coating. Lipid coating was performed as in [67]. Briefly, a gold electrode was cleaned in ammonium hydroxide: hydrogen peroxide: water (1:1:3 *v/v*) for 20–25 min at ca. 70 °C and washed in distilled water. Then, 2 μL of lipid solution in chlorophorm were deposited on the gold surface and air-dried. The value of the adsorbed mass was related to the resonator frequency by the relation:

Δ*F* = −*C* × Δ*m*,
(1)

where Δ*F* is the observed frequency change in Hz; Δ*m* is the change in mass per unit area, in μg/cm^2^; *C* is the sensitivity coefficient of the crystal (56.6 Hz × µg^–1^ × cm^2^). Lipid film was then hydrated, passing buffer 1 over the electrode surface until equilibrium was reached. Several hydration–dehydration cycles were performed to confirm that the lipids did not flake off. The studied solutions of Fb or RG in buffer 1 were degassed under vacuum and passed over the bare or lipid-coated electrode surface of a quartz resonator at a flow rate of 50 μL/min, according to vendor recommendations. The final concentrations of Fb and RG were 1 mg/mL. The adsorption of components on the electrode surface was judged from the change in the resonator oscillation frequency. All experiments were conducted at a temperature of 25 ± 0.05 °C and repeated at least three times.

### 4.6. Turbidimetry

The fibrin polymerization kinetics and lysis were studied by turbidimetry [68] using a Lambda 25 spectrophotometer (Perkin Elmer, Waltham, MA, USA). The initial Fb solution prepared in buffer 1 was mixed 1:1 with buffer 1 (control) or with solutions of RG or RG/DPPC in a spectrophotometer cuvette, incubated for 30 min at 25 °C, and Tb was added. The final concentrations of Fb and Tb were 1 mg/mL and 0.26 NIH U/mL, respectively. The final concentrations of DPPC—1 mg/mL. The final concentrations of RG were 1 and 5 mg/mL. The formation of a clot was registered by increasing the optical density at 350 nm. Fibrin lysis was initiated by spreading 10 μL of trypsin solution over the surface of the clot. The following parameters characterizing the kinetics of fibrin polymerization were measured: the lag period, the maximum rate of the reaction of fibrin formation and lysis, and the optical density when the kinetics reached a plateau [39]. Their values were calculated using the first derivative technique [69]. No model assumptions were made throughout the calculations.

### 4.7. Analysis of Gels and Their Components Interaction

(i)The resistance of gels to pH was studied by placing 1 mL of the formed gels without binase into a test tube with 10 mL of buffer 1 at pH 7.4 and pH 2 and measuring the optical density of supernatant at 280 nm after the incubation time.(ii)The interaction of lipid with binase was determined turbidimetrically measuring the half-transition temperature of a DPPC suspension with or without encapsulated binase. For these, an optical density at 350 nm was recorded upon continuous temperature scanning from 35 °C to 45 °C at a rate of 0.1 °C/min. The accuracy of temperature setting was not worse than 0.05 °C with a Pelkin Elmer Peltier thermostat. The dependence of optical density on temperature was approximated by the Vant-Hoff two-state model equation from the set of functions of the OriginPro 2015 package, obtaining the half-transition temperature.(iii)The release of binase from the composite gel was determined by subtracting the optical density in the supernatant at 280 nm in the control from the optical density in the experiment. To do this, 1 mL of the formed gel of various compositions with binase (experiment) and without binase (control) were placed in test tubes with 10 mL of buffer 1 at pH 7.4 and incubated for a specified time.(iv)Adsorption of Fb on lipid bilayers was measured using the microgravimetry technique.

### 4.8. Dynamic Light Scattering

Zeta potential and hydrodynamic particle diameter in the Fb/RG mixtures were measured on a Zetasaizer Nano spectrometer (Malvern Instruments Ltd., Malvern, UK). Low-salt buffer 2 was used to reduce Joule heating and electrode degradation in accordance with the manufacturer’s recommendations. The buffer was filtered through PTFE filters with a pore diameter of 0.22 µm. Fb solutions were prepared with a concentration of 0, 0.062, 0.125, 0.25, 0.5, 1 mg/mL and RG concentration 1 mg/mL.

### 4.9. IR Spectroscopy

IR spectra were recorded on an IR Affinity1 spectrophotometer (Shimadzu, Kyoto, Japan) using attenuated total internal reflection (ATR) attachment with a ZnSe crystal as a measuring element. The spectral resolution was 4 cm^−1^, and the number of accumulations was 512. The solutions of Fb and RG were successively passed through a flow microcell mounted on an ATR crystal for 1 h, alternating solutions of the studied substances and buffer 1. The spectrum for buffer 1 was registered first and used as background for all subsequent spectra, which allowed an automatic compensation for water and buffer salt contribution. Difference spectra were calculated subtracting the spectrum of pure adsorbed fibrinogen from the spectrum of fibrinogen with subsequently adsorbed RG. The spectrum of a pure RG solution was recorded separately, followed by washing, confirming a negligible adsorption of RG on the bare ATR surface. The cell was thermostated at 25 °C. The assignment of absorption bands in the IR spectra was based on the previous findings [36,70]. All spectral manipulations were performed with OPUS 7.0 (Bruker) software.

### 4.10. Scanning Electron Microscopy

For scanning electron microscopy (SEM), hydrogels were prepared by adding Tb to a Fb solution or to a Fb/RG mixture. The final concentrations of Tb, Fb and RG were 0.26 NIH U/mL, 1 mg/mL and 1 mg/mL, respectively. Fibrin gels were formed for 1.5 h at 22 °C and then washed three times with excess buffer for 15 min. The washed gels were fixed in 2% glutaraldehyde for 1 h, washed with water and dehydrated in increasing concentrations of ethanol. Samples were then immersed for 3 min in 100% hexamethyldisilazane (HMDS) and excess HMDS was removed on filter paper. The gels were studied using an Auriga Cross Beam scanning electron microscope (Carl Zeiss, Oberkochen, Germany).

### 4.11. Cytotoxicity towards Cancer Cells

Cytotoxicity of composite gels was analyzed using the MTT-test with 3-(4,5-dimethylthiazol-2-yl)-2,5-diphenyltetrazolium bromide (Sigma-Aldrich, St. Louis, MO, USA) following the manufacturer’s instructions. Duodenum adenocarcinoma HuTu-80 cells were obtained from the Russian cell culture collection of vertebrates (Saint Petersburg, Russia). Cells were cultured at 37 °C in a humidified atmosphere with 5% CO2 in DMEM (Gibco) supplemented by 10% fetal calf serum (HyClone, Logan, UT, USA), 100 U/mL penicillin, and 100 U/mL streptomycin. The experiment was carried out in 24-well plates, in which 1.5 × 10^5^ cells per well, containing 500 µL medium, were seeded 24 h before the experiment. After confluence, the medium was replaced with a fresh one, and cultural inserts (pore diameter 0.4 µm, SPL, Pocheon-si, Republic of Korea), containing composite gels directly formed in the inserts, were installed into the wells. The amount of binase in gel corresponded to the positive control concentration (130 µg of enzyme in 1 mL medium). After 24 h, the cell viability was measured.

### 4.12. Statistics

Statistical analysis and approximation of dependencies were performed using OriginPro 2015 (OriginLab Corp., Northampton, MA, USA). The average of three measurements and the standard deviation were determined. The statistically significant level was taken as *p* ≤ 0.05.

## 5. Conclusions

The present study demonstrates that it is possible to create composite protein-polysaccharide hydrogels based on non-degrading rhamnogalacturonan I (RG) and fibrin as a drug delivery vehicle for colon therapy. Despite the low gelling ability of RG alone, it interacts with fibrin to form a stable gel, resistant to both acid and neutral pH environments. The lipid particles filled with antitumor RNase binase were introduced to the forming gel and tested for binase release properties and its antitumor activities in vitro. The obtained results suggest the potential applicability of synthesized composite as a platform for protein drug delivery to the colon.

## Figures and Tables

**Figure 1 ijms-24-00926-f001:**
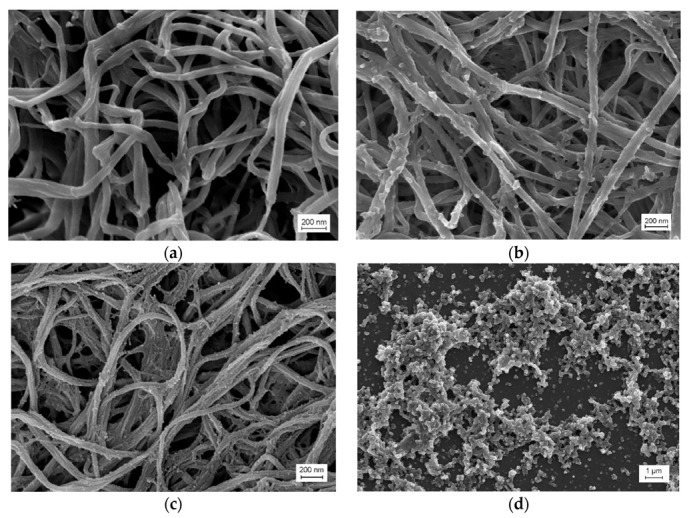
Micrographs of different gel samples obtained using scanning electron microscopy (SEM): Fibrin gel (**a**); composite gel Fibrin/RG (**b**); composite gel Fibrin/DPPC/RG with binase (**c**); pure RG (**d**). DPPC—1,2-dipalmitoyl-sn-glycero-3-phosphocholine.

**Figure 2 ijms-24-00926-f002:**
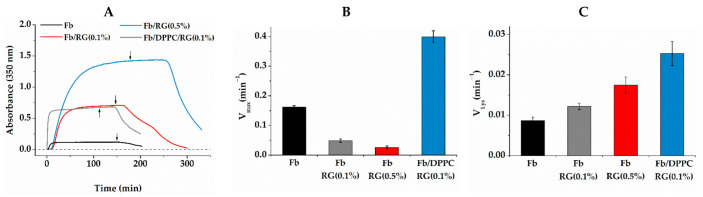
Kinetics of formation and lysis of fibrin (**A**) and its maximum rate of formation (**B**) and lysis (**C**) in various composite gels. RG concentration is denoted in parenthesis, Fb concentration is 1 mg/mL everywhere. The moment of trypsin introduction is marked with arrows (**A**).

**Figure 3 ijms-24-00926-f003:**
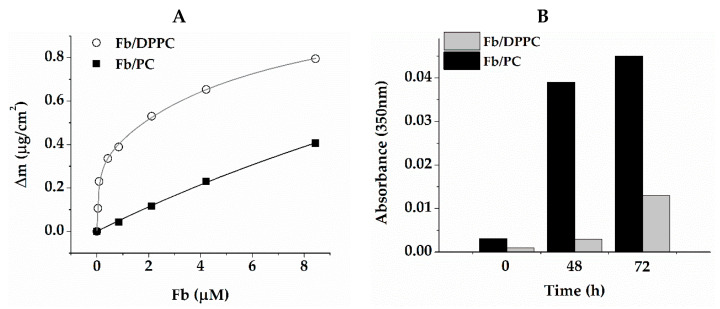
Mass of Fb adsorbed on lipid bilayer at increasing Fb concentration (**A**) and kinetics of lipid release during washing of fibrin–lipid gels (**B**). Data on (**A**) were approximated by two-stage binding kinetics: y = Bmax1 × x/(k1 + x) + Bmax2 × x/(k2 + x). The corresponding rate constants were: k1 = 0.063 ± 0.025, k2 = 6.34 ± 1.56 for DPPC and k1 = 0, k2 = 36.8 ± 8.6 for PC. PC—egg phosphocholine.

**Figure 4 ijms-24-00926-f004:**
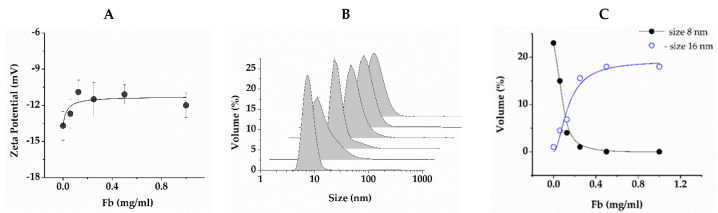
Dependence of zeta potential of RG at a concentration of 1 mg/mL (**A**), its hydrodynamic particle diameter (**B**) and proportion of particles with a size of 8 and 16 nm on concentration of Fb added (**C**). The solid line (**A**) is the approximation by formula y = B_max_ × x/(k + x) with the following parameters: Z-potential at adsorption limit B_max_ = −11.59 ± 0.4 mV/mg and affinity constant k = 0.03 ± 0.01 mg/mL. Fibrinogen concentration (**B**): 0, 0.062, 0.125, 0.25, 0.5 and 1.0 mg/mL. Blue line on (**C**) is the approximation by the Hill equation: y = Vol_max_ × x^n/(k^n + x^n), with Vol_max_ = 19.28 ± 1.97, k = 0.41 ± 0.08 mg/mL and n = 1.8 ± 0.4.

**Figure 5 ijms-24-00926-f005:**
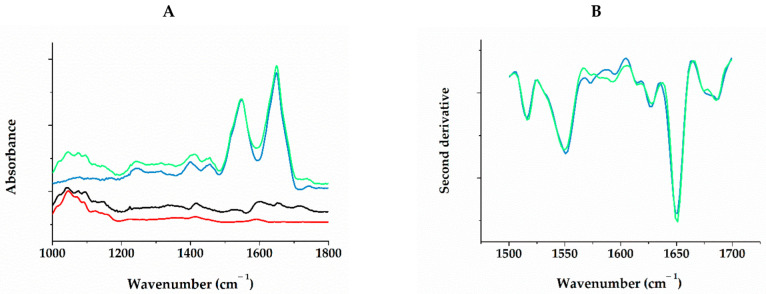
FTIR spectra of Fb adsorbed on the surface of ZnSe crystal (blue), Fb with RG adsorbed on top of it (green), difference between them (black) and RG spectrum (red) (**A**); and second derivatives of absorption spectra of Fb (blue) and Fb/RG complexes (green) in the region of peptide groups absorption (**B**).

**Figure 6 ijms-24-00926-f006:**
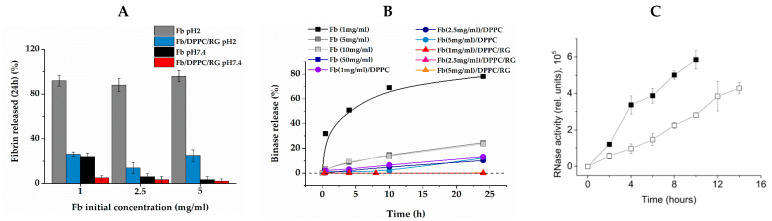
Release of fibrin (24 h of incubation, pH 2 and 7.4) (**A**); and dynamic of binase release (Tris–HCl pH 7.4) (**B**); from composite gels expressed as % of initial substances concentration in gel (Notice that data points for 1, 2.5 and 5 mg/mL Fb in composite Fb/DPPC/RG gels coincide). RG concentration is 1 mg/mL; (**C**) rise of binase activity in supernatant upon release from pure fibrin (closed symbols) and composite fibrin/DPPC (open symbols) gels (Tris–HCl pH 7.4, fibrin concentration 2.5 mg/mL).

**Figure 7 ijms-24-00926-f007:**
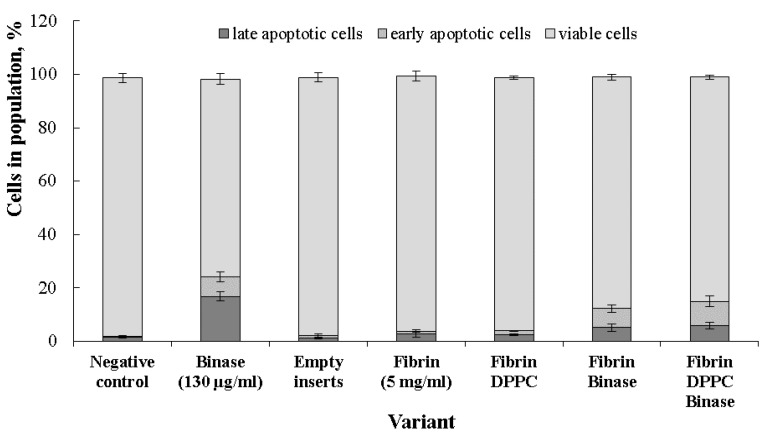
Apoptosis induction of *HuTu80* cells by pure substances (binase, fibrin) and three variants of composite gels after 24 h of monolayer treatment by substances released from inserts into cultivation medium with growing cells. Non-treated cells were taken as negative control.

**Table 1 ijms-24-00926-t001:** Half-transition temperature (Ttr) of turbidity in lipid suspensions.

Components	Ttr (°C)
DPPC	40.29 ± 0.05
DPPC + binase	39.45 ± 0.05

## Data Availability

Not applicable.

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
