# Peer review of "Fibrin-Rhamnogalacturonan I Composite Gel for Therapeutic Enzyme Delivery to Intestinal Tumors"

_ijms, 2023, doi:10.3390/ijms24020926_

Round 1

Reviewer 1 Report

Fibrin-rhamnogalacturonan I composite gel for therapeutic enzyme delivery to intestinal tumors, by Faizullin et al.

This manuscript reports the approach of delivering the RNAse binase using a hydrogel system consisting of fibrin/rhamnogalacturonan I/liposomes. The hydrogel is resistant to degradation in acidic environments (pH = 2) and slowly releases binase at pH = 7.4. These conditions are meant to simulate passage through the gastrointestinal tract, whereby the activity of binase is preserved in the acidic environment of the stomach and it is later released in the intestines with the aim of treating colorectal cancers.  The topic of the manuscript is within the scope of the Iternational Journal of Molecular Sciences. There are however, several items that need to be addressed because this manuscript can be considered publishable.

Some of the experimental descriptions raise a few questions:

-          Protein concentrations in DPPC liposome systems are determined using DPPC liposome suspensions as blanks. This approach would require careful control of DPPC concentration and liposome size distribution. How were these two later factors controlled? Maybe it is necessary to resort to an alternative method to quantitate protein such as labeling and/or the use of other colorimetric protein assays given that at 280 nm additional effects like scattering can interfere with the measurements. Also, in order for this blank subtraction method to work, it is necessary ensuring that the absolute absorbance of the systems remain within the linear range of Beer’s law (roughly absorbance < 1.5).

-          The microgravimetry experiments whereby Fb or RG are passed over a lipid coated QCM cyrtal requires a description of the coating procedure as well as a characterization of such surface modification.

-          The description of the interaction of lipid with binase involves a turbidity measurement. Yet, the conclusions reached are about a transition temperature. There is some ambiguity as to what is being measured. The turbidity measurement detects the stability of the liposome suspension, which is  not necessarily the same as the transition temperature of the lipid tails.

-          There are not enough details about the measurements of adsorption of Fb using the QCM technique ( e.g. buffers, concentrations, degassing of solutions, times, flow rates, etc.).

-          What is the rationale  for using two buffers  with different concentration and ionic strength (buffer 1 and buffer 2)?

-          Tris-HCl buffer cannot be used at pH = 2. This compound does not have buffering capability at such low pH. This implies poor control  of pH in the experiments intended to be made at pH = 2.

-          Measuring release of binase by using optical density at 280 nm is a poor choice. Given the presence of firbin and thrombin in the gels as well as the presence of liposomes it would be better to count with a method to directly measure binase activity. A measurement of O.D. at 280 nm does not necessarily means you are only measuring binase.

-          The description of statistics leaves much to be desired. Unfortunately, only the software used is mentioned and not the statistical techniques.

At several parts of the manuscript, proper references are not used (e.g. to back up the statement that 5 year survival rate is 11%).

At several parts of the manuscript there is use of scientific/technical terms that seem to be used very loosely. For example on line 91 “due to the hydrophobicity of galactan chains and their tendency to self-associate in an aqueous solution” . Given their large number of hydroxyl groups, why are they being characterized as hydrophobic? One would think there are plenty of groups to establish hydrogen bonds with water. And is the mechanism of self association really due to hydrophobic interactions?

Lines 92-93: “The coagulation ability of fibrinogen strongly depends on external conditions and on the state of fibrinogen molecule itself”. This is confusing and ambiguous. It is not clear what is meant by external conditions and what states the fibrinogen molecule can exist in other than its native conformation in solution.

Lines 130-131: “…slower diffusion of Fb monomers and protofibrils in a more viscous RG solution or an increase in the size of kinetic unit.” What is it meant by the size of kinetic unit? Or what is a kinetic unit? Did the authors bother to calculate the effect of viscosity on diffusion coefficients using Stokes-Einstein equation? Are those viscous effects large enough to explain the observations? Without such estimations this is just an invocation of scientific terms that may or may not be the real explanation for the observed phenomena.

Figures 2b and 2c. How are the Vmax and Vlys calculated? The curves clearly show that a single exponential model (first order kinetics) may not fit the entire data properly. It appears that some assumptions were made in order to make these calculations but those are not stated.

Lines 148-153: “Diffusion of protease trypsin into Fb/RG composite gel is slower, which manifests itself in the delay of gel disintegration after the moment of trypsin added. The delay depends on the RG concentration and increases rapidly in more concentrated RG solutions. However, the rate of lysis of fibers themselves increases in this case (Figure 2C), which can also be the result of modification of their internal structure.”  This statements make the assumption that the observed degradation of fibrin is solely due to the rate at which the fibers are degraded. This is not necessarily the case. Since this is measured by the turbidity and integrity of the gel, it is possible that different rates of degradation could be a reflection of differences in fibrin network structure (for example, degrees of branching, size of fibers and extent of crosslinking).

Lines 161 -162: “It follows from the presented data that the adsorption of Fb on the DPPC bilayer is a two stage process, while on the PC one it is a single-stage process”. This demands further explanation. What are those two stages of the process? What is the evidence for the existence of those stages? How was this conclusion reached? If it is because of the different shapes of the curves in Figue 3A it could simply mean differences in kinetics for the two processes. In either case, eventually the curves would be expected to reach a plateu after a sufficiently long period of time and they both would appear more or less linear for very short periods of time. Thus, the observed differences may simply reflect a process that quickly reaches the plateau value and a process that is far from reaching such plateau.

Figure 4a. Given the size of those error bars, is it really a trend? Are the differences statistically significant?

The equations described (Hill  equation) should be written with an equation editor. Also, too many significant figures are being reported for the fitting parameters.

Lines 210-214: “An increase in intensity in the 1715 cm-1 band indicates the protonation of a part of carboxyl groups. In this case, there is a corresponding decrease of the ionized carboxyl absorption at 1566 and 1399 cm-1. These carboxyl groups obviously belong to fibrinogen, since at frequencies of 1590 and 1415 cm-1, coinciding with the absorption maxima of the ionized RG uronic acid residues only an increase in intensity of difference spectrum is observed. ” A description of spectral manipulation and subtraction is lacking. It is not clear whether the authors removed the contributions from water that are always observed to overlap with the regions of the amide I band. It is not clear how the spectra were subtracted to avoid potential artifacts in their subtraction. Normally, there is a prominent band observed for proteins in the amide I region and the bands due to potential carboxyl groups in proteins are negligible. It is not clear that these different are not simple artifacts due to baseline drift. Given the presence of uronic acid groups in RG, it is not clear how the authors conclude such changes come from fibrinogen.

Line 237-238: “The interaction of binase with the DPPC bilayer was studied by changing the temperature of lipid phase transition”. It appears that the authors meant to say that the interaction of binase with the DPPC bilayer changes the transition temperature and such change was measured. Not that this interaction was measured by, somehow, changing the transition temperature. How this was done is not clearly described in the experimental section. If anything, the experimental section mentions a turbidity measurement as a function of temperature. The later  does not necessarily measure transition temperatures of the lipid chains but changes in the stability of the liposome suspension.

Table 1. Too many significant figures and most thermometers lack the accuracy and precision to confidently measure changes in temperature as small as those reported.

Lines 264-266: “At 264 pH 7.4, the destruction of pure fibrin sharply slows down, which is due to increased electrostatic interactions between fibrin monomers [53].” This is another instance where the authors throw scientific jargon without reflecting on whether their statement is true or not. Given the isoelectric point of fibrinogen, it is expected to have a negative charge at pH 7.4 (same for RG). This would create electrostatic repulsion. It is not clear how electrostatic repulsion can explain stability of fibrin. Given that there are other intermolecular forces in effect, how do the authors conclude that electrostatic interactions are responsible for the observed phenomenon? Did they even test varying ionic strengths in order to modulate the extent of such electrostatic interactions? This explanation does not make sense.

Author Response

Dear Reviewer,

thank you for your attempts to make our work better and your useful questions and comments. We tried to correct our article according to your comments. Our answers and corrections in the text are shown in blue.

Some of the experimental descriptions raise a few questions:

-          Protein concentrations in DPPC liposome systems are determined using DPPC liposome suspensions as blanks. This approach would require careful control of DPPC concentration and liposome size distribution. How were these two later factors controlled? Maybe it is necessary to resort to an alternative method to quantitate protein such as labeling and/or the use of other colorimetric protein assays given that at 280 nm additional effects like scattering can interfere with the measurements. Also, in order for this blank subtraction method to work, it is necessary ensuring that the absolute absorbance of the systems remain within the linear range of Beer’s law (roughly absorbance < 1.5).

Answer: Due to enzyme deficiency, we found it unacceptable to use staining or covalent labeling methods to measure the binase content in liposomes, since these treatments do not allow protein recovery. We were aware that liposomes scattering creates certain problems in the measuring of protein content at 280 nm. For this, only freshly prepared suspensions were used. Care was taken to ensure that the total absorbance at 280 nm due to liposome scattering and protein absorption did not exceed 0.5 in both the measurement and reference cuvettes. Besides, the actual content of binase in liposomes was verified by measuring the protein concentration in the washing fractions. The results obtained by the two methods agreed with each other within 10%.

-          The microgravimetry experiments whereby Fb or RG are passed over a lipid coated QCM cyrtal requires a description of the coating procedure as well as a characterization of such surface modification.

Answer. We introduced a description of lipid coating procedure in the relevant section at page 12.

-          There are not enough details about the measurements of adsorption of Fb using the QCM technique ( e.g. buffers, concentrations, degassing of solutions, times, flow rates, etc.).

Answer. We have added the missing descriptions.

-          The description of the interaction of lipid with binase involves a turbidity measurement. Yet, the conclusions reached are about a transition temperature. There is some ambiguity as to what is being measured. The turbidity measurement detects the stability of the liposome suspension, which is not necessarily the same as the transition temperature of the lipid tails.

Answer: We agree that the term "transition temperature" is used somewhat loosely, despite this term has been used in the past literature (see, for example, Eker F. et al. Application of turbidity technique on peptide-lipid and drug-lipid interactions. Journal of Molecular Structure 482–483 (1999) 693–697). Be that as it may, the change in the stability of the lipid dispersion verifies the interaction of liposomes with the ligand.

-          What is the rationale for using two buffers with different concentration and ionic strength (buffer 1 and buffer 2)?

Answer. Low-salt buffer 2 was used in the Zeta-potential measurements to reduce the Joule heating and electrode degradation in accordance with the manufacturer's recommendations.

-          Tris-HCl buffer cannot be used at pH = 2. This compound does not have buffering capability at such low pH. This implies poor control of pH in the experiments intended to be made at pH = 2.

Answer. When studying the stability of fibrin gels, the same buffer composition was used at pH 7 and pH 2 only for the constancy of the salt composition of the medium. The pH 2 value was controlled manually during the experiment. No pH deviations of more than ±0.2 units were found.

-          Measuring release of binase by using optical density at 280 nm is a poor choice. Given the presence of firbin and thrombin in the gels as well as the presence of liposomes it would be better to count with a method to directly measure binase activity. A measurement of O.D. at 280 nm does not necessarily means you are only measuring binase.

Answer. We indeed performed the activity measurements of binase immobilized either on pure fibrin or enclosed in DPPC liposomes, embedded in fibrin. No significant differences were found between activity and 280 nm release kinetics.

-          The description of statistics leaves much to be desired. Unfortunately, only the software used is mentioned and not the statistical techniques.

Answer. We have added the missing information to the relevant section.

At several parts of the manuscript, proper references are not used (e.g. to back up the statement that 5 year survival rate is 11%).

Answer. We carefully checked the reference list and replaced several items to more relevant or recent ones.

At several parts of the manuscript there is use of scientific/technical terms that seem to be used very loosely. For example on line 81 “due to the hydrophobicity of galactan chains and their tendency to self-associate in an aqueous solution” . Given their large number of hydroxyl groups, why are they being characterized as hydrophobic? One would think there are plenty of groups to establish hydrogen bonds with water. And is the mechanism of self association really due to hydrophobic interactions?

Answer. We apologize for the inaccuracy. It would be more correct to speak about the significantly lower hydrophilicity of galactan chains compared to other side groups of rhamnogalacturonans. As revealed in Ref. 34 – 36, the increased tendency of galactan-enriched potato rhamnogalacturonan I to self-associate is due to reduced hydration of side chains, stiffness of enzymatically modified backbone with cleaved repeated acid groups and complex interactions between side chains. The corrections are made in the text.

Lines 92-93: “The coagulation ability of fibrinogen strongly depends on external conditions and on the state of fibrinogen molecule itself”. This is confusing and ambiguous. It is not clear what is meant by external conditions and what states the fibrinogen molecule can exist in other than its native conformation in solution.

Answer. To large extent, the fibrin clot structure is kinetically controlled. Numerous factors, influencing clotting kinetics were outlined in refs. 16, 38 – 41, 46. Besides the obvious environmental factors, such as salinity, pH, temperature etc, the chemical modifications (for example, deglycolization and point mutations) or even conformational alterations upon adsorption on surfaces or crowding would significantly affect clotting.  We would like to avoid cumbersome enumerations, however, following your suggestions we have added some clarifications to this part of the text.

 Lines 130-131: “…slower diffusion of Fb monomers and protofibrils in a more viscous RG solution or an increase in the size of kinetic unit.” What is it meant by the size of kinetic unit? Or what is a kinetic unit? Did the authors bother to calculate the effect of viscosity on diffusion coefficients using Stokes-Einstein equation? Are those viscous effects large enough to explain the observations? Without such estimations this is just an invocation of scientific terms that may or may not be the real explanation for the observed phenomena.

Answer. We carefully analyzed the reviewer’s objections. Despite the existence of some arguments in favor of reciprocal dependence of diffusion coefficient on polymer concentration (see, for example, [Matthias Krüge and Markus Rauscher. Diffusion of a sphere in a dilute solution of polymer coils // J. Chem. Phys. 131, 094902 (2009)]), there may be various and unknown factors complicating such a behavior in fibrinogen solutions. We generally agree with reviewer’s criticism and have rewritten the paragraph.

Figures 2b and 2c. How are the Vmax and Vlys calculated? The curves clearly show that a single exponential model (first order kinetics) may not fit the entire data properly. It appears that some assumptions were made in order to make these calculations but those are not stated.

Answer. The definitions of lag time, Vmax and Vlys may be found elsewhere (see, for example [38]). Their values were calculated numerically using the first derivative technique which is widely described in literature (see, for example, [Tutwiler V, Litvinov RI, Lozhkin AP, et al. Kinetics and mechanics of clot contraction are governed by the molecular and cellular composition of the blood. Blood. 2016;127(1):149-159. doi:10.1182/blood-2015-05-647560]). No model assumptions were made throughout the calculations.

Lines 148-153: “Diffusion of protease trypsin into Fb/RG composite gel is slower, which manifests itself in the delay of gel disintegration after the moment of trypsin added. The delay depends on the RG concentration and increases rapidly in more concentrated RG solutions. However, the rate of lysis of fibers themselves increases in this case (Figure 2C), which can also be the result of modification of their internal structure.”  This statements make the assumption that the observed degradation of fibrin is solely due to the rate at which the fibers are degraded. This is not necessarily the case. Since this is measured by the turbidity and integrity of the gel, it is possible that different rates of degradation could be a reflection of differences in fibrin network structure (for example, degrees of branching, size of fibers and extent of crosslinking).

Answer. We fully agree with the reviewer’s comment and have made the appropriate changes in the paragraph.

Lines 161 -162: “It follows from the presented data that the adsorption of Fb on the DPPC bilayer is a two stage process, while on the PC one it is a single-stage process”. This demands further explanation. What are those two stages of the process? What is the evidence for the existence of those stages? How was this conclusion reached? If it is because of the different shapes of the curves in Figue 3A it could simply mean differences in kinetics for the two processes. In either case, eventually the curves would be expected to reach a plateu after a sufficiently long period of time and they both would appear more or less linear for very short periods of time. Thus, the observed differences may simply reflect a process that quickly reaches the plateau value and a process that is far from reaching such plateau.

Answer. Of course, it may be that, but within the measurement error the data are most well approximated by two-stage binding for DPPC and only one-stage – for PC. We added the relevant explanations in the text.

Figure 4a. Given the size of those error bars, is it really a trend? Are the differences statistically significant?

Answer. The large error bars result from continuous drift due to Joule heating during the measurement process. Varying the measurement time the overall trend has been verified.

The equations described (Hill  equation) should be written with an equation editor. Also, too many significant figures are being reported for the fitting parameters.

Answer. We made the relevant corrections.

Lines 210-214: “An increase in intensity in the 1715 cm-1 band indicates the protonation of a part of carboxyl groups. In this case, there is a corresponding decrease of the ionized carboxyl absorption at 1566 and 1399 cm-1. These carboxyl groups obviously belong to fibrinogen, since at frequencies of 1590 and 1415 cm-1, coinciding with the absorption maxima of the ionized RG uronic acid residues only an increase in intensity of difference spectrum is observed. ” A description of spectral manipulation and subtraction is lacking. It is not clear whether the authors removed the contributions from water that are always observed to overlap with the regions of the amide I band. It is not clear how the spectra were subtracted to avoid potential artifacts in their subtraction. Normally, there is a prominent band observed for proteins in the amide I region and the bands due to potential carboxyl groups in proteins are negligible. It is not clear that these different are not simple artifacts due to baseline drift. Given the presence of uronic acid groups in RG, it is not clear how the authors conclude such changes come from fibrinogen.

Answer. We introduced the detailed description of the FTIR experiment in the relevant section. Concerning the assignment of carboxyl band changes to fibrinogen, we again emphasize that those band positions in fibrinogen and RG are clearly distinct which makes it possible to distinguish them unambiguously.

Line 237-238: “The interaction of binase with the DPPC bilayer was studied by changing the temperature of lipid phase transition”. It appears that the authors meant to say that the interaction of binase with the DPPC bilayer changes the transition temperature and such change was measured. Not that this interaction was measured by, somehow, changing the transition temperature. How this was done is not clearly described in the experimental section. If anything, the experimental section mentions a turbidity measurement as a function of temperature. The later  does not necessarily measure transition temperatures of the lipid chains but changes in the stability of the liposome suspension.

Answer. We added to the Methods (section Analysis of gels and their components interaction) the description of the fitting model. The reported difference in the half-transition temperatures is statistically significant.

Table 1. Too many significant figures and most thermometers lack the accuracy and precision to confidently measure changes in temperature as small as those reported.

Answer. The cited errors of transition temperature refer to the approximation model and are within the thermometer accuracy which is better than one decimal place. Nevertheless, we corrected the figures in the Table.

Lines 264-266: “At 264 pH 7.4, the destruction of pure fibrin sharply slows down, which is due to increased electrostatic interactions between fibrin monomers [53].” This is another statement is true or not. Given the isoelectric point of fibrinogen, it is expected to have a negative charge at pH 7.4 (same for RG). This would create electrostatic repulsion. It is not clear how electrostatic repulsion can explain stability of fibrin. Given that there are other intermolecular forces in effect, how do the authors conclude that electrostatic interactions are responsible for the observed phenomenon? Did they even test varying ionic strengths in order to modulate the extent of such electrostatic interactions? This explanation does not make sense.

Answer. We apologize for a loosely used terminology. It is established in the literature that fibrin monomers at physiological pH aggregate to the fibrils due to point-to-point local charge interactions between “knobs” and “holes” demasked upon thrombin cleavage of fibrinogen. Are there other interactions and of what strength is still the subject of discussion, but just the neutralization of these local charges at acidic pH is crucial for fibrin disaggregation [N. Okumura, O. V. Gorkun, S. T. Lord. Severely impaired polymerization of recombinant fibrinogen // J Biol Chem 1997;272(47):29596-601] and repeated in [Weisel JW, Litvinov RI. Fibrin Formation, Structure and Properties. Subcell Biochem. 2017;82:405-456.]. We have tried to formulate this statement clearer in the revised text.

Reviewer 2 Report

1-      It would be better if the authors can add some numerical results to the abstract for better comparison and understanding.

2-      Usually, the number of keywords is 5, please reduce the number of keywords and mention the most important ones.

3-      The authors explained the results before material and methods section, so, it is necessary to add or mention abbreviation section before results. What is DPPC referred to?

4-      The authors mentioned that ternary system of Fb/RG/DPPC has formed thinner fiber but there is no number (size) to compare with pure fiber.

5-      What is the reason that by increasing the concentration of fibrin, the release rate decrease?

6-      As it can be seen from the toxicity chart, the proportion of non-apoptotic cells was 291 86.6±1.07% and 84±0.8% for Fb and Fb/DPPC, respectively with a low amount of apoptotic cells. Do the authors think that it is enough for treatment of cancer? How do the authors explain that? Is there a reference?

7-      It would be better that the authors measure the toxicity of the prepared structure to the healthy human cells to understand if they are safe to be used in body or not,

Author Response

Dear Reviewer,

thank you for your attempts to make our work better and your useful questions and comments. We tried to correct our article according to your comments. Our answers and corrections in the text are shown in blue.

Comments and Suggestions for Authors

  • It would be better if the authors can add some numerical results to the abstract for better comparison and understanding.

Answer: Thank you for your preposition, but we suppose that our abstract is most suitable to present the main points of the work.

  • Usually, the number of keywords is 5, please reduce the number of keywords and mention the most important ones.

Answer. We reduced the number of keywords.

  • The authors explained the results before material and methods section, so, it is necessary to add or mention abbreviation section before results. What is DPPC referred to?

Answer. We added the missed abbreviations.

  • The authors mentioned that ternary system of Fb/RG/DPPC has formed thinner fiber but there is no number (size) to compare with pure fiber.

Answer. We added the relevant figures to the text.

  • What is the reason that by increasing the concentration of fibrin, the release rate decrease?

Answer. To our opinion, the more dense mesh is realized with increasing fibrin concentration which retards diffusion of single enzyme and especially the liposome-embedded ones.

  • As it can be seen from the toxicity chart, the proportion of non-apoptotic cells was 291 86.6±1.07% and 84±0.8% for Fb and Fb/DPPC, respectively with a low amount of apoptotic cells. Do the authors think that it is enough for treatment of cancer? How do the authors explain that? Is there a reference?

Answer. The Figure 7 shows the apoptotic effect in 24 hours, with increasing time the number of apoptotic cells will increase. The absolute value of apoptosis level is not important in this case; it was important to show that the immobilized enzyme does not lose its pro-apoptotic effect compared to the pure enzyme.

  • It would be better that the authors measure the toxicity of the prepared structure to the healthy human cells to understand if they are safe to be used in body or not

Answer. We have previously shown that binase does not act on normal fibroblasts but has a toxic effect on ras-transformed tumor cells (Ilinskaya O, Decker K, Koschinski A, Dreyer F, Repp H. Bacillus intermedius ribonuclease as inhibitor of cell proliferation and membrane current. Toxicology. 2001 Jan 2;156(2-3):101-7). The other work (Cabrera Fuentes H.A., Zelenikhin P.V., Kolpakov A.I., Preissner K.T., Ilinskaya O.N. Comparative toxicity of binase towards tumor and normal cells. Uchenye Zapiski Kazanskogo Universiteta, 2010, v.152, No 3, p.143-148) characterizes cytotoxic activity of binase towards solid tumor cells: pulmonary adenocarcinoma A549, human fibrosarcoma HT1080 and murine glioma C6. The enzyme possesses high cytotoxic effect on A549 and C6 cells and does not at the same time inhibit proliferation of HT1080 cells. This fact could be explained by the different expression levels of ras-oncogenes by the tested cell lines. RNase did not show cytotoxicity towards human umbilical vein endothelial cells (HUVEC) in concentration range of 0.1–300 µg/ml. Note that the safety of RNases application has been confirmed: for example, RNase is used as a commercial antiviral drug registered in the Vidal reference book. (https://www.vidal.ru/drugs/ribonuclease__24326).

Concerning the fabricated composite gel, the absence of a toxic effect on normal cells of its ingredients - fibrin and rhamnogalacturonan I (potato galactan) as well as DPPC liposomes - was previously shown [Khotimchenko M. Pectin polymers for colon-targeted antitumor drug delivery. Int J Biol Macromol. 2020 May 5:S0141-8130(20)33147-0. doi: 10.1016/j.ijbiomac.2020.05.002;

Heher, P.; Mühleder, S.; Mittermayr, R.; Redl, H.; Slezak, P. Fibrin-Based Delivery Strategies for Acute and Chronic Wound Healing. Advanced Drug Delivery Reviews 2018, 129, 134–147, doi:10.1016/j.addr.2017.12.007;

Wang, S.-S.; Yang, M.-C.; Chung, T.-W. Liposomes/Chitosan Scaffold/Human Fibrin Gel Composite Systems for Delivering Hydrophilic Drugs—Release Behaviors of Tirofiban In Vitro. Drug Delivery 2008, 15, 149–157, doi:10.1080/10717540801952456 ].

Round 2

Reviewer 2 Report

The authors addressed most of the comments and can be published.